# Minimal clinically-important differences for the 'Liverpool Osteoarthritis in Dogs' (LOAD) and the 'Canine Orthopedic Index' (COI) client-reported outcomes measures

**John F. Innes**[1,2]*, **Mark A. Morton**[3,4], **B. Duncan X. Lascelles**[5,6,7,8]

**1** Movement Veterinary Referrals, Preston Brook, Runcorn, Cheshire, United Kingdom, **2** School of Veterinary Science, University of Liverpool, Leahurst Campus, Neston, Liverpool, United Kingdom, **3** ChesterGates Veterinary Specialists, Chester, United Kingdom, **4** Canine Cruciate Registry, RCVS Knowledge, The Cursitor, London, United Kingdom, **5** Translational Research in Pain (TRiP) Program, Department of Clinical Sciences, College of Veterinary Medicine, North Carolina State University, Raleigh, North Carolina, United States of America, **6** Comparative Pain Research and Education Center, College of Veterinary Medicine, North Carolina State University, Raleigh, North Carolina, United States of America, **7** Thurston Arthritis Center, UNC School of Medicine, Chapel Hill, North Carolina, United States of America, **8** Center for Translational Pain Research, Department of Anesthesiology, Duke University, Durham, North Carolina, United States of America

* j.f.innes@liverpool.ac.uk

**Data Availability Statement:** The data underlying the results presented in the study are available from RCVS Knowledge Canine Cruciate Registry (https://caninecruciateregistry.org/). Access to data

## Abstract

Client-reported outcomes measures (CROMs) have been previously validated for the evaluation of canine osteoarthritis. A published systematic review indicated that the 'Liverpool Osteoarthritis in Dogs' (LOAD) and the 'Canine Orthopedic Index' (COI) can be recommended for use in dogs with osteoarthritis; these CROMs have also been used in the context of measuring surgical outcomes of dogs with orthopaedic conditions. However, the minimal clinically-important differences (MCIDs) for these CROMs have not been investigated. Such estimates would be useful for investigators and regulators so that these CROMs can be used in clinical trials. Data from the RCVS Knowledge Canine Cruciate Registry were extracted, and baseline and 6 week follow-up data on dogs that had received surgery for cranial cruciate ligament rupture were used to make estimates of MCIDs using distribution-based and anchor-based methods. Data from 125 dogs were categorised based on the anchor question and LOAD and COI scores analysed accordingly. The four anchor-based methods provided a range of MCIDs for each CROM (1 to 8.8 for LOAD and 3.5 to 17.6 for COI). In the two different distribution-based methods, the MCIDs for LOAD ranged from 1.5 (effect size) to 2.4 (standard error of measurement) and the effect size method yielded a result of 2.2 for COI. The results showed that the value of the MCIDs depended on the method that was applied. Receiver operator characteristic curves provided areas under the curve (AUCs) greater than 0.7, which indicated that the cut-off point was acceptable; LOAD had the greater AUC at 0.867. In summary, the authors currently recommend a MCID of '4' for LOAD and '14' for COI although further work in other clinical contexts (such as osteoarthritis associated with chronic pain) is required to add confidence to these estimates. For the first time, we have provided estimates for MCIDs for these two CROMs

can be requested by sending and email to RCVS Knowledge (info@rcvsknowledge.org) or writing to: RCVS Knowledge, The Cursitor, 38 Chancery Lane, London WC2A 1EN, UK. The authors of this study had no special access privileges that others would not have".

**Funding:** The authors received no specific funding for this work.

**Competing interests:** I have read the journal's policy and the authors of this manuscript have the following competing interests: John Innes is co-holder, with University of Liverpool, of the licence for the LOAD client-reported outcomes measure. LOAD is licensed to Elanco Animal Health.

which will facilitate sample size estimates in future clinical studies that use these CROMs as outcomes measures.

## Introduction

Measuring outcomes in canine joint disease is challenging and can involve objective measures, such as gait analysis, or semi-objective measures such as client or veterinarian-based structured questionnaires. Whilst gait analysis is objective, accurate and reliable [1], it is expensive, time-consuming and not widely available. Clinical assessments by veterinarians are of limited value in that many of the clinical signs of joint disease are not apparent in an in-clinic setting and such assessments can suffer from a floor effect [2]. Assessments by clients is appealing because clients spend considerable time with their dogs but it is important for client-based assessment tools to be thoroughly tested so that one can have confidence in the results and understand the limitations of such measures. Unfortunately, in the past, researchers in veterinary orthopaedics often used unvalidated questionnaires. However, in the last two decades several client-reported outcomes measures (CROMs) have been developed and validation data published [3–15].

In human medicine, patient-reported outcomes measures (PROMs) are widely accepted as tools to assess outcomes across a wide range of health conditions, including joint diseases. PROMs are used by regulatory bodies, researchers and healthcare providers to assess efficacy and quality of care [16].

In veterinary medicine, the client can be used as a proxy to communicate their interpretation of their animal's health status [6, 7, 17]. Client-reported outcomes measures have also been called 'owner-reported outcomes measures' (OROMs) [15] or 'clinical metrology instruments' [8, 14]. Several CROMs have been developed for canine joint disease and some of these have been through validation processes. Basic validation involves assessment of 'face', 'criterion' and 'construct' validity, reliability, responsiveness and practicality.

The COSMIN initiative (COnsensus-Based Standards for the selection of health Measurement INstruments) aims to improve the selection of health measurement instruments, both in research and clinical practice, by developing tools for selecting the most suitable instrument for a given situation. COSMIN (www.cosmin.nl) is an international initiative consisting of a multidisciplinary team of researchers with expertise in epidemiology, psychometrics, and qualitative research, and in the development and evaluation of outcome measurement instruments in the field of healthcare, as well as in performing systematic reviews of outcome measurement instruments. The COSMIN initiative has published a comprehensive guideline for systematic review of PROMs as well as a checklist for assessing bias.

A COSMIN-based systematic review of CROMs in canine joint disease was recently published [15]. Seventeen publications describing the validation of 6 CROMs were selected and evaluated with the COSMIN. Of the 6 CROMs evaluated, the authors concluded that three [Liverpool Osteoarthritis in Dogs (LOAD), Canine Orthopedic Index (COI) and the Canine Brief Pain Inventory (CBPI)] provided evidence of sufficient content validity. The authors further concluded that LOAD, COI and CBPI can be recommended for use in dogs with osteoarthritis.

Beyond validity, there are other features of a CROM which are useful to know. The minimal clinically-important difference (MCID) is useful for the purposes of study design and sample size estimates in research and clinical trials. Regulators may use MCID to define the threshold

between 'responder' and 'non-responder' in regulatory clinical trials. In addition, MCID is useful in the context of monitoring patients' responses to interventions and in clinical-decision-making. MCID is the smallest change in the score of an outcome measure that a patient or client would identify as important [18]. MCID can be estimated using anchor-based methods or distribution-based methods [19]. Distribution-based methods rely on the statistical characteristics of a group's baseline PROM/CROM scores to determine how much of a change may be clinically important. In comparison, anchor-based methods compare the change in clients' CROM scores to how clients score on a second, explicit metric of client opinion of change (the anchor question).

One of the most common joint diseases in dogs is associated with rupture of the cranial cruciate ligament rupture (CCLR) and the subsequent secondary osteoarthritis. The RCVS Knowledge 'Canine Cruciate Registry' (CCR) was launched in 2021 with the aim of understanding the efficacy and safety of surgical interventions for CCLR. The registry is web-based and participation is free for both owners and veterinary surgeons in the UK. Veterinary surgeons in the UK that are performing surgery for ruptured cranial cruciate ligament in dogs are eligible to contribute to the registry. Prior to surgery, the registry collects information on signalment and disease status of patients, along with baseline LOAD and COI scores. Surgical interventions are recorded as well as follow up LOAD, COI and anchor questions at set time intervals following surgery.

In this study, we set out to estimate MCIDs for LOAD and COI using data from the RCVS Knowledge 'Canine Cruciate Registry' (www.caninecruciateregistry.org).

## Materials and methods

The RCVS Knowledge CCR has ethical approval from the RCVS Ethics Review Panel (ref 2020-14-Morton). Participating veterinary surgeons obtained informed consent from clients to register their dog on the CCR. At registration, a more detailed consent to participation was obtained electronically, along with consent for the client's e.mail address to be stored as well as consent to regular contact. After registration, and prior to surgery, clients provided information on signalment and disease status of patients and completed baseline LOAD and COI scores. Immediately after surgery, participating veterinary surgeons were requested to confirm the owner's submission regarding the dog (breed, sex, age, bodyweight) as well as provide the clinical details of the disease (joint affected, status of cruciate ligament, status of menisci) and surgery (technique used, implants used). Completion of the surgical report triggered automated emails to be sent to the client at set time intervals after surgery. These emails requested that the client complete online versions of LOAD, COI and an anchor question. The anchor question was, "Compared to before surgery, how is your dog now?". The possible responses to the anchor question were, "much better", "somewhat better", "the same", "somewhat worse" and "much worse". Only the pre-surgery and 6 weeks post-operative data were used in this study. Any patients without baseline scores, 6 week scores and 6 week anchor question answers were excluded.

Data from the CCR were exported on 22 June 2022 on to an Excel (Microsoft, Seattle, USA) spreadsheet. Statistical analyses were performed with Graphpad Prism 9 (San Diego, USA). Only data from clients that had answered the anchor question at 6 weeks were included in the analysis. Based on the anchor question responses at 6 weeks, two groups were defined–owners who answered 'the same' and owners who answered 'somewhat better'. The Mann Whitney U test was used to compare baseline characteristics of the 'the same' and 'somewhat better' groups. For categorical data, Fisher's exact test was used. The Wilcoxon signed-rank test was used to evaluate change in scores over time (pre-surgery to 6 weeks post-surgery) for each of

the CROMs. The Mann-Whitney U test was used to compare differences in the CROMs and the mean change in the CROMs between the "the same" and "somewhat better" groups. The level of significance was set *a priori* at p < 0.05. Because this study was only concerned with estimating MCID for LOAD and COI using longitudinal anchor-based methods and distribution-based methods, we did not analyse clinical outcomes *per se* and we did not analyse complications, nor other variables that may have affected the outcome.

Four anchor-based methods were used to calculate the MCID. The 'average change' (AC) corresponded to the mean change in the score of the 'somewhat better' group. The 'minimum detectable change' (MDC) approach defined minimal change as the smallest change that can be considered beyond the measurement error with a given level of confidence (95%). Therefore, in this context of a positive change in LOAD and COI being a reduction in total score, MCID was equal to the lower value of the 95% confidence interval for the average change in score that is seen in the no change ("the same") group. The 'change difference' was defined as the difference in the average change in score between the "somewhat better" and "the same" groups. A receiver operating characteristic (ROC) curve was used to define the cut-off point that best discriminated between the minimal change and no change groups. The optimal cut-off point was estimated using the point that maximized both specificity and sensitivity (i.e. where Youden's index was highest). The area under the ROC curve (AUC) was calculated to assess reliability. An AUC value of 0.7 to 0.8 was considered acceptable and an AUC value of 0.8 to 0.9 was considered excellent.

We also undertook two distribution method estimates of MCID. The first was based on the effect size. The effect size is calculated as the difference in a group's mean score from pre-treatment to post-treatment divided by the standard deviation (SD) of the pre-treatment scores. An effect size of 0.2 is considered small and so MCID can be estimated by calculating ($SD_{pretreatment}$ *0.2) [19]. Alternatively, although the use of SD for distribution-based calculation of MCID is well established, a criticism is that SD is a property of the cohort being studied so the resultant MCID calculation may be sample-dependent. In contrast to SD, the 'standard error of measurement' (SEM) of CROM scores is independent of the patient cohort being studied and an intrinsic property of the CROM assessment tool [19]. SEM was estimated by:

$$SEM = SD \times \sqrt{(1 - r)}$$

Where *r* is the reliability of the instrument. For LOAD, as a value for 'r' we used a previously published estimate of reliability from a test-retest scenario [8]. Although test-retest reliability for COI has been estimated, a correlation coefficient for *r* was not available, only *kappa* values for interrater agreement [4] and so it was not possible to estimate MCID using this method.

## Results

### Study population

Data from 125 subjects from the CCR fulfilled the inclusion criteria. Two subjects had complete LOAD data but incomplete COI data and were, accordingly, excluded from COI analysis.

Of the 125 dogs, breed was identified for 79. Thirty-two different breed types were represented with the Labrador Retriever (7) and mixed breed (6) being the most frequently represented. Sixty-three dogs were male and 62 were female. Of these, 58 and 55, respectively, were neutered. The mean age was 7 years (SD = 3.0).

The surgical procedure performed was identified for 119 of 125 dogs. Cranial closing wedge osteotomy was performed in 26 dogs, tibial plateau levelling osteotomy in 73 dogs and tibial tuberosity advancement in 20 dogs.

**Table 1. Mean (SD) pre-operative and post-operative scores for 'Liverpool Osteoarthritis in Dogs' (LOAD) and 'Canine Orthopedic Index' (COI).**

| CROM | n | Pre-operative score | 6 weeks score | p value[#] |
|------|---|---------------------|---------------|-----------|
| LOAD | 125 | 19 (± 7) | 12 (± 6) | <0.0001 |
| COI | 123 | 32 (± 11) | 12 (± 10) | <0.0001 |

[#] Wilcoxon signed rank test.

The mean pre-operative and 6-week scores for LOAD and COI are shown in Table 1. Both CROMs demonstrated a significant difference between pre-operative and post-operative scores. For the anchor question, there were 95 dogs in the "much better" group, 18 in the "somewhat better", nine in the "the same" group (one of these had missing COI data), one in the "somewhat worse" group and two in the "much worse" group. The mean (SD) changes in LOAD and COI from baseline to 6 weeks for these five groups are illustrated in Fig 1. There was only one response in the "somewhat worse" group and the CROM scores for this dog were at odds with the response to the anchor question in that both CROM scores decreased by 10 and 18 points for LOAD and COI respectively. Accordingly, this response was considered an anomaly and this group was not used in any further analysis.

## MCID estimates

Of the clients who responded that their dogs were 'the same' at 6 weeks post-operatively compared to before surgery, nine and eight completed a LOAD and COI respectively. For both instruments, there were 18 clients who responded to say that their dogs were "somewhat better" at 6 weeks post-operatively. Pre-operative, 6 week and score changes for these two groups are summarised in Table 2.

The MCID estimates with the four anchor-based methods and the two distribution-based methods are shown in Table 3. The four anchor-based methods provided a range of MCIDs for each CROM (1 to 8.8 for LOAD and 3.5 to 17.6 for COI). In the two different distribution-based methods, the MCID for LOAD ranged from 1.5 (effect size) to 2.4 (SEM) and the effect size method yielded a result of 2.2 for COI. The results showed that the value of the MCID depended on the method that was applied. Both AUCs that were defined by the ROC curve were greater than 0.7, which indicated that the cut-off point was acceptable; LOAD had the greater AUC at 0.867.

## Discussion

This study estimated MCIDs for two validated CROMs for use in canine joint disease. We used data from the CCR at baseline and at the 6 weeks timepoint not because 6 weeks is a useful sampling point to assess outcomes following surgery for CCLR but because we expected a spread of responses to the anchor question which would generate sufficient data to allow for MCID estimates. It is important to note this clinical context and to understand that these estimates of MCIDs may not translate to other clinical contexts such as medical management of chronic musculoskeletal pain. It is recognised that MCID estimates for PROMs can vary with the clinical context [20].

The study population was drawn from the RCVS Knowledge CCR, the first registry for canine cruciate ligament surgery. This is the first analysis using CCR data. The analysis has shown that omissions in data entry are relatively common and this may inform future policies with respect to data entry (e.g. more mandatory fields). Nevertheless, the data were sufficient

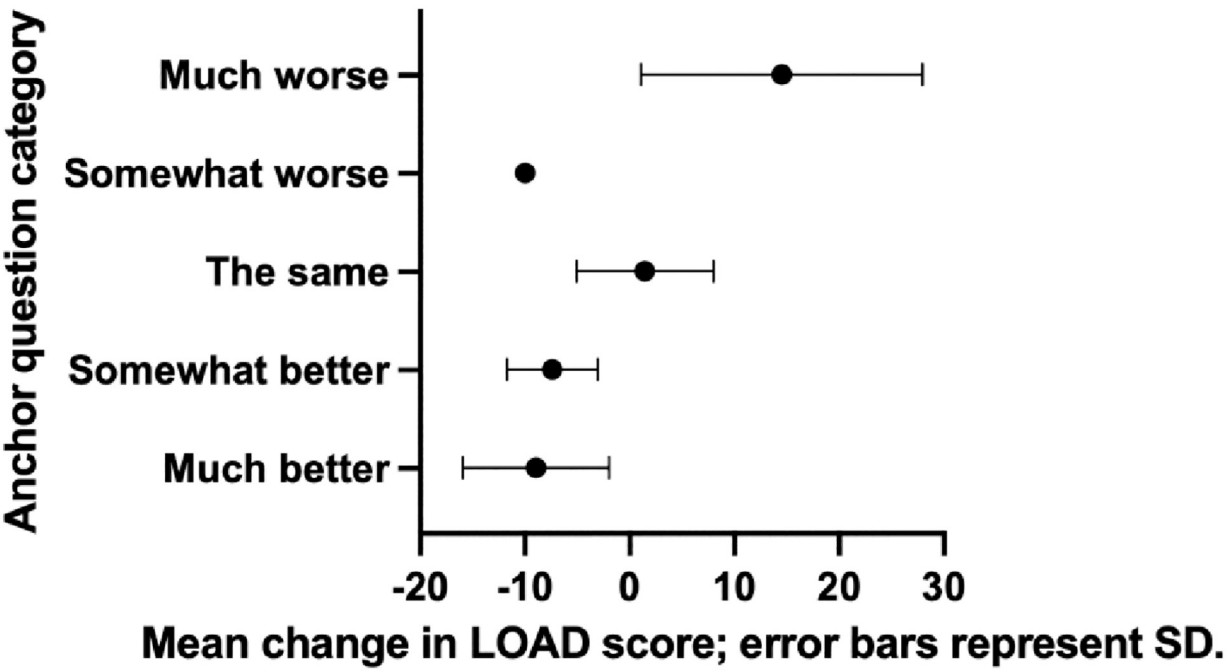

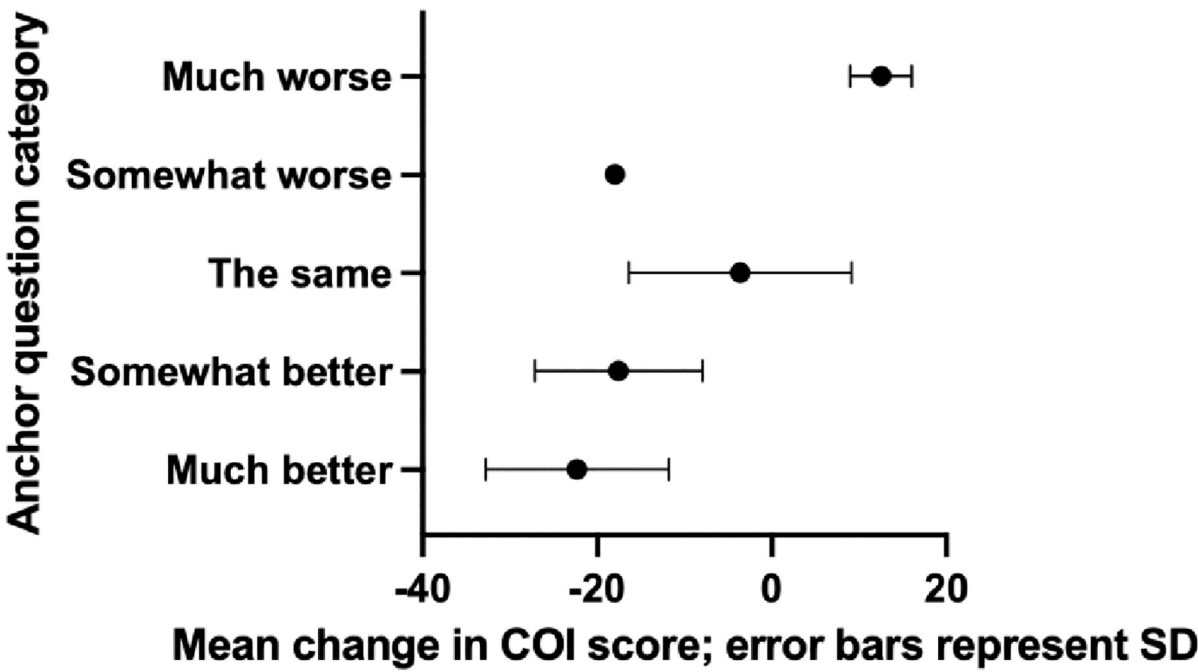

**Fig 1. Mean change in LOAD and COI scores at six weeks according to the anchor question response category.** Error bars represent 95% standard deviations.

**Table 2. Pre- and Postoperative CROMs in the 'the same' group and the 'somewhat better' group.**

| CROM | 'The same' group | 'Somewhat better' group | P value |
|---|---|---|---|
| LOAD | | | |
| n | 9 | 18 | |
| Pre | 16.9 ± 8.8 | 20.6 ± 6.2 | 0.2763 |
| Post | 18.3 ± 7.3 | 13.2 ± 4.5 | 0.0803 |
| Mean score change | 1.4 ± 6.5 | -7.4 ± 4.3 | 0.0033 |
| COI | | | |
| n | 8 | 18 | |
| Pre | 24.4 ± 12.7 | 33.5 ± 10.3 | 0.1011 |
| Post | 20.8 ± 10.0 | 15.9 ± 7.3 | 0.2479 |
| Mean score change | -3.6 ± 12.7 | -17.6 ± 9.6 | 0.0188 |

There was not any significant difference between the two groups in terms of baseline characteristics (bodyweight, age, sex, surgical technique). There was not any significant difference in LOAD and COI scores between these two groups at baseline. For both LOAD and COI, there was a significant difference between these two groups at 6 weeks in terms of change in score (p<0.0001, Wilcoxon signed rank test), and so we were able to continue with MCID estimates for both instruments.

for us to use them to estimate MCID for LOAD and COI. With 125 dogs included, the baseline mean (SD) LOAD and COI scores documented here will be useful to future investigators and regulators; the baseline values for LOAD are similar to those previously published for dogs with CCLR [21, 22].

Although both LOAD and COI were able to show a significant change in scores for dogs between baseline and 6 weeks, the scores at 6 weeks should not be interpreted as typical clinical outcomes for CCR surgery because 6 weeks is too early to assess outcomes after such surgery. In support of this, the mean CROMs scores at 6 weeks reported here are higher than previously reported outcomes for CCLR surgeries after longer post-operative periods [21–24]

Estimates of MCID may be affected by the context of the estimate (extrinsic factors) and by the method of calculation (intrinsic factors). For example, it is possible that post-operative instructions to clients and management of the patients may have impacted on LOAD or COI scores because, at 6 weeks after surgery, a dog's exercise may be limited by the client. The MCID values reported here provide an estimate of the change in LOAD and COI that could be considered clinically meaningful for dogs with cruciate ligament disease treated surgically. While these MCID values may prove to be generally reflective of a range of joint disease and treatments, it will also be important to estimate MCID for LOAD and COI in other clinical contexts, such as medical management of osteoarthritis and other surgical interventions. That

**Table 3. MCIDs for LOAD and COI.**

| Anchor-based | | | | | Distribution-based | |
|---|---|---|---|---|---|---|
| CROM | AC | CD | MDC | ROC Curve (AUC) | Effect size | SEM |
| LOAD | -7.4 | -8.8 | -3.6 | -1.0 (0.867) | ±1.5 | ±2.4 |
| COI | -17.6 | -13.9 | -14.3 | -3.5 (0.785) | ±2.2 | n/a |

CROM, client-reported outcomes measure; AC, average change; CD, change difference; MDC, minimal detectable change; ROC, receiver operator characteristic; AUC, area under ROC curve; SEM, standard error of measurement; LOAD, Liverpool Osteoarthritis in Dogs; COI, canine orthopedic index. For anchor-based MCID estimates, values are minus because it was a reduction in score that was measured as improvement. For distribution-based methods, the estimate is positive or negative.

said, these initial estimates will provide investigators with initial estimated values that can be used to inform sample size estimates when using these CROMs in clinical studies and clinical trials. In addition, the values may also be used by regulators to set thresholds for categorising subjects as responders or non-responders. However, it will be preferable to have MCID estimates for clinical contexts such as medical management of osteoarthritis.

The anchor-based and distribution-based methods generated different values for MCID. Although there is no consensus as to the best method to estimate MCID, it has been argued that anchor-based estimates are more clinically-relevant. However, distribution-based methods can be used to support anchor-based estimates, or used in the absence of anchor-based methods [25]. In addition, the distribution methods are based on the baseline variability in CROM scores and hence stem from a larger data set (125 and 123 subjects for LOAD and COI respectively in this study) compared to the anchor-based methods. When comparing the two distribution methods there is some argument to prefer the SEM method because, in contrast to effect size which is a factor of SD, according to classical test theory, the SEM of CROM scores is independent of the patient cohort being studied and an intrinsic property of the CROM assessment tool [19].

We used four different anchor-based methods because previous studies in human patients have shown that different methods generate different values for MCID [26, 27]. For LOAD, 'change difference' generated the largest estimate of MCID at 8.8, and for COI the largest estimate was for 'average change' at 17.6. For both LOAD and COI, the ROC curve method generated the smallest MCID estimate of the anchor-based methods and these were broadly similar to the distribution-based method estimates.

The AUC for the ROC curve analysis indicated that both LOAD and COI had acceptable ability to discriminate between dogs that were "somewhat better" compared to dogs that were "the same". The AUC can be considered an index of discriminating ability of a test and although LOAD had a higher AUC than COI, suggesting it is better able to discriminate between the two groups, the relatively small sample sizes would limit the confidence in this difference and further data are required.

In summary, it seems reasonable to suggest that, using the MDC method, and acknowledging that a LOAD score must be a whole integer, a 'working' MCID of '-4' (rounded from -3.6) for LOAD is currently appropriate. Using such a threshold would be supported by the ROC (-1.0) anchor-based method and both distribution-based methods (ES ±1.5 and SEM ±2.4). For COI, using a similar approach would suggest that a 'working' MCID of '-14' (rounded from -14.3) is reasonable and this would be supported by AC (-13.9), ROC (-3.5) and ES (±2.2).

Because CROMs are completed by clients, CROMs have an inherent risk of a caregiver placebo effect (CPE). A CPE was demonstrated when clients completed a questionnaire regarding their dog's lameness [28] and the data were compared to ground reaction forces as measured by a force platform, although this questionnaire was not a validated CROM. Nevertheless, a CPE was also suggested for a validated CROM, the Canine Brief Pain Inventory (CBPI), when compared to activity monitor counts [29] although activity monitors are less validated than force platforms as an objective outcomes measure in veterinary orthopaedics. Interestingly, in a trial of anti-nerve growth factor antibody versus placebo for the treatment of canine osteoarthritis, the CBPI showed a significant CPE whereas the LOAD did not [30] suggesting that not all CROMs are equally susceptible to CPE. This may be because LOAD predominantly asks clients to assess 'activity/exercise' and 'stiffness/lameness' whereas CBPI predominantly asks clients to assess 'pain' [7] which is perhaps a more nebulous concept for clients to score. To support that concept, LOAD has consistently shown a weak correlation with objective assessment of load bearing [7, 31] whereas CBPI has been more varied in that regard [13, 31]. Recent data for COI also indicate criterion validity versus objective assessment of load bearing [31].

That all said, in this study, a CPE could have influenced our data set, although evaluation of the anchor-based categories as shown in Fig 1 indicates that 'the same' group had 'change in LOAD' scores with a mean just above zero and 'change in COI' scores just below zero indicating consistency between the CROMs data and the clients' overall impression. However, the CPE remains a factor that must be considered when using CROMs, particularly in open-label settings [32] and likely even more so if the treatment is invasive [33]. The authors recommend blinding and placebo control if CROMs are to be used as the primary measure and it is noteworthy that regulators have taken a similar view [34, 35]. In human medicine, when blinding is not possible, it is recommended that PROMs collection should be combined with functional, imaging and biochemical biomarkers adding an element of objectivity [36].

There are other features of a CROM that we did not address in this study and should be the focus of future work. We did not estimate the client-acceptable clinical state (CACS) which is a threshold in the postoperative score where a client is likely to define the outcome as 'satisfactory'. This would be the equivalent of the patient acceptable symptom state (PASS) for a PROM in human medicine [37]. The CACS threshold would be defined as the postoperative CROM score that was predictive of client satisfaction. Neither did we estimate values for 'substantial clinical benefit' (SCB) which would be defined as the clinical value that the client considers as 'substantial improvement'. The SCBs would be calculated as the cut-off point that best discriminates between the substantial ("much better") and non-substantial ("somewhat better," "about the same," or "somewhat worse") improvement groups. For both of these parameters, we felt they would be better calculated after a longer post-operative period (e.g., 6 months).

## Conclusions

For the first time, we present estimates of MCIDs for LOAD and COI and conclude working values of '-4' and '-14' respectively. These estimates can be used by researchers wishing to use these CROMs in future studies and will be useful to clinicians monitoring patients. Further work is required to provide further MCID estimates in different clinical contexts such as osteoarthritis of the hip and elbow, and to estimate the additional parameters of 'client-acceptable clinical state' and 'substantial clinical benefit'.

## Acknowledgments

The authors are grateful to RCVS Knowledge for access to the RCVS Knowledge CCR and to all veterinary surgeons and clients participating in the registry. We are also grateful to Elanco Animal Health Ltd for use of the LOAD instrument which is exclusively licensed from University of Liverpool. The authors thank Nichola Archer-Thompson for helpful comments on the manuscript and Rhiannon Hornett for guidance on data extraction from the CCR.

## Author Contributions

**Conceptualization:** John F. Innes.

**Data curation:** Mark A. Morton.

**Formal analysis:** John F. Innes.

**Investigation:** John F. Innes, B. Duncan X. Lascelles.

**Methodology:** John F. Innes, Mark A. Morton, B. Duncan X. Lascelles.

**Project administration:** John F. Innes.

Writing – **original draft:** John F. Innes.

Writing – **review & editing:** Mark A. Morton, B. Duncan X. Lascelles.

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
