## [Decision Letter · Decision Letter 0]

1 Nov 2022

PONE-D-22-28671Minimal clinically-important differences for the ‘Liverpool Osteoarthritis in Dogs’ (LOAD) and the ‘Canine Orthopedic Index’ (COI) client-reported outcomes measuresPLOS ONE

Dear Dr. Innes,

Thank you for submitting your manuscript to PLOS ONE. After careful consideration, we feel that it has merit but does not fully meet PLOS ONE’s publication criteria as it currently stands. Therefore, we invite you to submit a revised version of the manuscript that addresses the points raised during the review process.

We look forward to receiving your revised manuscript.

Kind regards,

Richard Evans

Academic Editor

PLOS ONE

Journal Requirements:

"I have read the journal's policy and the authors of this manuscript have the following competing interests: John Innes is co-holder, with University of Liverpool, of the licence for the LOAD client-reported outcomes measure. LOAD is licensed to Elanco Animal Health."

Additional Editor Comments:

Great paper, please see the reviewers' comment--they look reasonable. 

Reviewers' comments:

Reviewer's Responses to Questions

**Comments to the Author**

1. Is the manuscript technically sound, and do the data support the conclusions?

Reviewer #1: Yes

Reviewer #2: Yes

2. Has the statistical analysis been performed appropriately and rigorously? 

Reviewer #1: Yes

Reviewer #2: Yes

3. Have the authors made all data underlying the findings in their manuscript fully available?

Reviewer #1: Yes

Reviewer #2: Yes

4. Is the manuscript presented in an intelligible fashion and written in standard English?

Reviewer #1: Yes

Reviewer #2: Yes

5. Review Comments to the Author

Reviewer #1: This is a very well written paper addressing an important question in veterinary medicine, what is the minimal clinically-important difference in CROMs. The primary revision that is required in this manuscript is rewording of certain areas (e.g. abstract, discussion) to make it clear to the reader that the MCID identified addresses short-term postoperative recovery/acute pain after knee surgery. It does not address chronic pain associated with osteoarthritis, which these outcome measures are commonly used for.

Line 31, 41, 281, 283, 333-Be specific what you studied and/or have identified (read above paragraph). I do not see how these numbers could be translated to chronic OA studies or perioperative pain studies. If you disagree, please justify in the discussion how/why these findings can be translated to other clinical situations.

Line 48-While some authors have done this it is not recommended. Guidelines have been published. It may be worthwhile rewording this sentence so future investigators don’t simply dismiss gait analysis but instead perform it correctly.

Line 65-Please reword to, “…proxy to communicate their interpretation of their animal’s…”

Reviewer #2: Major points

This is a well-written and much-needed article. I have one primary concern.

1. The caregiver placebo effect. The principles used in this manuscript come from human healthcare. If a human has less pain because of a placebo effect, then the human still has less pain and clinically that is what matters. But if a veterinary client incorrectly perceives that a dog is in less pain, that is, there is a caregiver placebo effect, then the dog is still in pain. In this manuscript, the authors found MCID for the caregiver placebo effect, not for a real reduction in pain. In other words, I could design a study MCID calculation study just like this one, but with the dogs like the ones that Conzemius and Evans used, which had no objective reduction in lameness but the owners perceived a reduction in lameness. I could have the owners fill out the LOAD and do all the same calculations in this manuscript, and find an MCID, and it would all be fiction.

3. My question is, to what degree are your MCIDs affected by caregiver placebo effect? If that is not measurable, then I suggest the authors note my point in the discussion section to whatever degree they feel appropriate.

Very minor point

I don't think that PLOS articles are checked for grammar or style by PLOS. The authors wrote "e.mail." Should it be e.mail or email? Most style guides prefer email. Fewer prefer e-mail. I can't find any that prefer e.mail.

6. PLOS authors have the option to publish the peer review history of their article (what does this mean?). If published, this will include your full peer review and any attached files.

Reviewer #1: No

Reviewer #2: No

---

## [Author Response · Author response to Decision Letter 0]

20 Dec 2022

Response to reviewers document uploaded

---

## [Editor Report · Decision Letter 1]

11 Jan 2023

Minimal clinically-important differences for the ‘Liverpool Osteoarthritis in Dogs’ (LOAD) and the ‘Canine Orthopedic Index’ (COI) client-reported outcomes measures

PONE-D-22-28671R1

Dear Dr. Innes,

We’re pleased to inform you that your manuscript has been judged scientifically suitable for publication and will be formally accepted for publication once it meets all outstanding technical requirements.

Kind regards,

Richard Evans

Academic Editor

PLOS ONE

Additional Editor Comments:

Thank you for your thorough responses to the reviewers' suggestions.

---

## [Editor Report · Acceptance letter]

23 Jan 2023

PONE-D-22-28671R1 

Minimal clinically-important differences for the ‘Liverpool Osteoarthritis in Dogs’ (LOAD) and the ‘Canine Orthopedic Index’ (COI) client-reported outcomes measures 

Dear Dr. Innes:

I'm pleased to inform you that your manuscript has been deemed suitable for publication in PLOS ONE. Congratulations! Your manuscript is now with our production department. 

Kind regards, 

on behalf of

Dr. Richard Evans 

Academic Editor

PLOS ONE